# Difficulty with mobility among the aged in Ghana: Evidence from Wave 2 of the World Health Organization's Study on Global Ageing and Adult Health

**Kingsley Boakye**[1,2]*, **Antoinette Ama Aidoo**[1,3], **Mohammed Aliyu**[1], **Daniel Boateng**[1,2], **Emmanuel Kweku Nakua**[1]

1 Department of Epidemiology and Biostatistics, School of Public Health, Kwame Nkrumah University of Science and Technology, Kumasi, Ghana, 2 Department of Global Public Health and Bioethics, Julius Center for Health Sciences and Primary Care, University Medical Center, Utrecht University, Utrecht, The Netherlands, 3 St. Patrick's Nursing and Midwifery Training College, Offinso, Ashanti Region, Ghana

* boakyekingsley9452@gmail.com

**Data Availability Statement:** The dataset used in this study, WHO's SAGE Wave 2, is not publicly available. However, it can be accessed through the WHO website at https://apps.who.int/healthinfo/

## Abstract

### Background

Globally, the population is rapidly ageing, stemming from a recent decline in mortality, and an increase in life expectancy. About 727 million people globally were aged $\geq$65 in 2020, and 1 in 6 people will be $\geq$65 years by 2050. About 7% of Ghana's population was over 60 years in 2010, and projected to be 12% by 2050. However, the aged are confronted with degenerative conditions that translate into difficulty with mobility. The study was conducted to investigate the difficulty with mobility among the aged in Ghana.

### Methods

The study utilised a cross-sectional dataset of the 2014/2015 (wave 2) Study on Global Ageing and Adult Health and included 1,856 participants aged $\geq$50 years. The survey command was applied to adjust for sampling biases and the design of the study. At 5% alpha level, a chi-square test of independence was conducted to determine the association between dependent and independent variables. At 95% confidence interval and 5% alpha level, three-level multilevel logistic regression models were performed. The fixed-effects were presented in odds ratio and the random effects were presented using the Intra-Class Correlation. All analysis were performed using STATA statistical software version 16.0.

### Results

Out of the 1,856 participants, 40.3% had difficulty with mobility. Additionally, age (80 and above) [AOR = 3.05, 95%CI = 1.78–5.22], self-reported poor/bad health status [AOR = 2.39, 95%CI = 1.35–4.23], having severe/extreme difficulty performing household activities [AOR = 25.12, 95%CI = 11.49–44.91], experienced severe/extreme bodily pains [AOR = 4.56, 95%CI = 2.16–9.64], severe/extreme sleep problems [AOR = 4.15, 95%CI = 1.68–

systems/surveydata/index.php/catalog/?page=
1&ps=15 or requested via email at
sagesurvey@who.int. The authors did not have any
special access privileges to this data.

**Funding:** The authors received no specific funding
for this work.

**Competing interests:** The authors have declared
that no competing interests exist

10.29], and participants with difficulty with sight/vision [AOR = 1.56, 95%CI = 1.16–2.10] had higher odds of difficulty with mobility.

## Conclusion

The aged in Ghana had relatively higher prevalence (40.3%) of difficulty with mobility which is influenced by demographics, health status, and degeneration associated with ageing. This highlights the need to provide social support and strengthen social capital for the aged in Ghana, especially those with difficulty with vision, experiencing bodily pains and had poor health status. Additionally, the Government and stakeholders should provide assistive devices for the aged and geriatric care including recreational fields and care homes to address the health and physical needs of the aged in Ghana.

## Background

The global population is swiftly ageing and this is attributed to significant reductions in mortality at younger ages in low and middle-income countries (LMICs) and continuing increases in life expectancy among the aged globally [1, 2]. In the year 2020, an estimated 727 million people were over 65 years and by 2050, an estimated 20% of the global population will be over 65 years [3]. According to WHO, about 2 billion people globally will be over 60 years by 2050 and over 400 million will be aged 80 years and above, out of which 80% will dwell in LMICs [4]. This trend in ageing will increase following the scientific, medical and technological advancements globally [5].

Ageing is inevitable, however, rapid ageing is problematic for people in LMICs [6] and this requires critical attention [7]. Unfortunately, the world is unable to address the swift demands arising from the growing numbers of aged, even in higher-income countries [6]. In sub-Saharan Africa (SSA), including Ghana, the rapid ageing population is presenting challenges to the already weakened health systems. By 2050, SSA is projected to have 8.3% of its population as aged [4]. Ghana had 7% of the aged (60+ years) in 2010, a figure among the highest proportions in SSA, and projected to increase to 12% by 2050 [8]. As of 1960, life expectancy in Ghana was 46.9 years, however, in 2020, it increased to 63.7 years which is further projected to increase to 70 years by 2050, and 76.2 years by 2100 [9]. Meanwhile, this increase in life expectancy is not tantamount to an increase in good health [10]. This is a clear-cut indication of the impending burden of care for the aged.

The inadequate preparedness to deal with this demographic shift, coupled with little attention given to the aged is quite problematic [11]. Ageing brings about a deterioration in health leading to high levels of morbidity and difficulty with mobility, termed as impaired movement [11]. Mobility is key to living independently and quality of life. Mobility also accounts for the well-being of the aged population because it promotes healthy living [12]. However, mobility limitations are increasingly prevalent among the aged, especially those aged 70 years and above, affecting more than one-third (35%) [13]. The difficulty in mobility among the aged contributes to increased falls, hospitalisation, mortality and decreased quality of life [13]. Hence, as the population ages, maintaining independent mobility is critical, especially for women who mostly have heightened risk of functional decline and disability [14]. The difficulty with mobility among aged could result in loss of muscle mass (sarcopenia), osteoporosis and obesity and it is associated with health problems and injuries [11]. The aged with mobility

difficulties have increased rates of morbidity, poorer quality of life and are more probable to be socially isolated [15]. Hence, it is important to pay critical attention to the situation of the aged, particularly in areas of research and policy [11]. There is currently a paucity of literature in Ghana regarding mobility among the aged. This lack of data means there will be inaccurate and unreliable data for policymaking, apt interventions and formulation of policies and programs in relation to mobility [16]. The aim of this paper was to investigate difficulty with mobility among the aged in Ghana. Thus, the paper provides current evidence and contributes to bridging the literature gap on difficulty with mobility among the aged in Ghana.

## Methods

### Study design and sampling

This study analysed data from 2014/2015 (Wave 2) Study on Global Ageing and Adult Health (SAGE) [16]. In brief, SAGE is a multi-country study that collects data to complement existing ageing data sources to inform policy and programmes. The study employed multi-stage cluster sampling techniques where clusters were systematically sampled with known non-zero selection probability and households residing in the selected clusters identified/listed and individuals in those selected households interviewed for the study The WHO and the University of Ghana Medical School, Department of Community Health collaborated to implement SAGE Wave 2 study. Detailed description of the methods is published elsewhere [16].

The Wave 2 WHO's SAGE study interviewed 4,735 individuals, primarily focusing on older adults ($\geq$50 years), but for comparison purposes, a smaller sample of those aged 18 to 49 years was also included in the study [17]. However, this study was restricted to 1,856 aged (50+ years) who had complete information on variables of interest.

### Dependent variable

The main dependent variable was difficulty with mobility which was computed from a question from the questionnaire of the WHO SAGE study; "overall in the last 30 days, how much difficulty did you have with moving around?". This question had responses "none", "mild", "moderate", "severe" and "extreme/cannot do". The response categories, "mild", "moderate", "severe" and "extreme/cannot do" were recoded as "difficulty", and "none" was classified as "no difficulty". "No difficulty" and "difficulty" were then recoded into "0" and "1", respectively. This categorization was adopted to determine difficulty with mobility rather than the degree of difficulty. This dichotomizing approach of difficulty with mobility was adopted from a previous study [9].

### Independent variables

The study considered 20 independent variables to help determine factors associated with difficulty with mobility among the aged in Ghana. The variables were socio-demographic factors (age, residence, gender, marital status, formal education, ethnicity, religion, work experience and Body Mass Index (BMI)), lifestyle (ever drink alcohol, smoke tobacco, engage in vigorous activity) and health status (self-reported health, bodily pains, sleep problem, diagnosed with arthritis, depression, and difficulty with vision, performing household activities and ever had road traffic accident). These variables have been used to determine difficulty with mobility among the aged elsewhere [18]. The initial coding for these selected variables in the WHO's SAGE study is attached as a supplementary file (S1 Table).

## Ethical clearance

The authors of this manuscript did not participate in the actual data gathering processes. Hence, we sought permission by officially writing for the dataset. However, the WHO's Wave 2 study sought ethical approval from the World Health Organization's Ethical Review Board (reference number RPC149) and the Ethical and Protocol Review Committee, College of Health Sciences, University of Ghana, Accra, Ghana. The SAGE study followed all ethical procedures and ensured that participants' rights were not violated. Written informed consent was obtained from all study respondents and the dataset was anonymised before making it available to the public.

## Data management and analytical procedures

Data for the study was downloaded after the authors sought for permission. The data was then cleaned using self-written commands to check for incompleteness and errors in STATA version 16.0. All errors including completeness and consistency were checked before actual analysis.

Data cleaning using self-written commands as well as statistical analysis were performed using STATA statistical software version 16.0 [19]. First, "survey set command" was applied to account for the sampling biases, complex survey design and generalisability of findings, respectively. Following that, descriptive computations were conducted to describe the general sampled characteristics. At 5% alpha threshold, a chi-square test of independence was conducted to ascertain the association between dependent and independent variables. Collinearity diagnostics was performed and reported using variance inflation factor (VIF) with a cut-off of 10. The results indicated no evidence of collinearity between independent variables (Mean VIF = 1.38, Maximum VIF = 1.64, Minimum VIF = 1.15) (see S2 Table).

A multilevel logistic regression model was conducted to determine the association between the dependent and independent variables. The extension from the single-level model to multilevel regression model is warranted due to the clustering and/or hierarchical structure of the SAGE dataset, where individuals are nested within households/EAs [17]. The presence of nesting leads to correlation between observations in each cluster and violation of the independence assumption in Generalised Linear Models (GLM) [20]. Hence, the use of a single-level model (GLM) will lead to misspecification and errors in parameter estimates [21] making multilevel the suitable and excellent model. The study considered individuals (ids) as level 1 identifiers and EAs as the level-2 identifier to explain the variability and clustering between and within-groups [21].

As such, at 95% confidence interval (95% CI) and 5% alpha threshold, three-stage multilevel regression models were built to assess the relationship between dependent and independent variables. The first was the null model (Model 0) which accounted for the variability of the outcome variables that could be attributed to the clustering of the PSUs/clusters. Subsequently, a second model (Model I) also assessed the crude association between dependent and independent variables. Finally, Model II (adjusted model) was conducted to account for the effect of other covariates. The outputs were presented using Odds Ratio (OR) and 95% CI. Due to the nesting, the Akaike Information Criterion (AIC) was utilised to assess the model fitness and the model with low AIC values was selected as the best model [22].

## Results

Table 1 presents the socio-demographic characteristics of the 1,856 study participants. Descriptively, 458 (39.6%) were aged between 50–59 years, 1,091 (52.9%) reside in rural areas, and 958 (56.1%) were males. Nine hundred and fifty-eight (60.2%) were married, 976 (61.0%)

**Table 1. Socio-demographic characteristics of respondents.**

| Variable | Unweighted | | Weighted | |
|---|---|---|---|---|
| | Frequency (n = 1,856) | Percentage (%) | Frequency (n = 1,856) | Percentage (%) |
| **Age group [in years]** | | | | |
| 50–59 | 458 | 24.7 | 458 | 39.6 |
| 60–69 | 680 | 36.6 | 680 | 31.8 |
| 70–79 | 485 | 26.1 | 485 | 19.1 |
| ≥80 | 233 | 12.6 | 233 | 9.5 |
| **Place of residence** | | | | |
| Urban | 765 | 41.2 | 765 | 47.1 |
| Rural | 1,091 | 58.8 | 1,091 | 52.9 |
| **Gender** | | | | |
| Male | 958 | 51.6 | 958 | 56.1 |
| Female | 898 | 48.4 | 898 | 43.9 |
| **Marital status** | | | | |
| Never married | 62 | 3.3 | 62 | 3.0 |
| Married | 985 | 53.1 | 958 | 60.2 |
| Separated/divorced | 234 | 12.6 | 234 | 11.8 |
| Widowed | 575 | 31.0 | 575 | 25.0 |
| **Ever had formal education** | | | | |
| No | 880 | 47.4 | 880 | 38.9 |
| Yes | 976 | 52.6 | 976 | 61.1 |
| **If yes, highest educational level (n = 976)** | | | | |
| Basic | 434 | 44.5 | 434 | 46.6 |
| Secondary | 473 | 48.5 | 473 | 46.3 |
| Tertiary | 69 | 7.0 | 69 | 7.1 |
| **Ethnicity** | | | | |
| Akan | 937 | 50.5 | 937 | 46.9 |
| Ewe | 131 | 7.1 | 131 | 11.0 |
| Ga-adangbe | 215 | 11.6 | 215 | 13.1 |
| Gruma/grusi/guan | 129 | 6.9 | 129 | 6.2 |
| Mande-busanga/mole-dagbani | 444 | 23.9 | 444 | 22.8 |
| **Religion** | | | | |
| No religion | 62 | 3.4 | 62 | 3.0 |
| Christian | 1,359 | 73.2 | 1,359 | 75.3 |
| Moslem | 328 | 17.7 | 329 | 17.1 |
| Traditionalist | 106 | 5.7 | 106 | 4.6 |

had formal education, 937 (46.9%) belonged to the Akan ethnic group, and 1,359 (75.3%) were Christians.

## Difficulty with mobility among the aged in Ghana

Fig 1 is a pictorial presentation of the weighted prevalence of difficulty with mobility among the aged in Ghana. More than half of the participants (59.7%) had no difficulty and 747 (40.3%) had difficulty with mobility.

The results of the chi-square test of independence showed that age group, gender, marital status, ever attended school, health status, difficulty house activities, experience bodily pains, sleep problems, visual difficulty, engage in vigorous activities and ever had road traffic accident were associated with difficulty with mobility among the aged at 5% alpha level (Table 2).

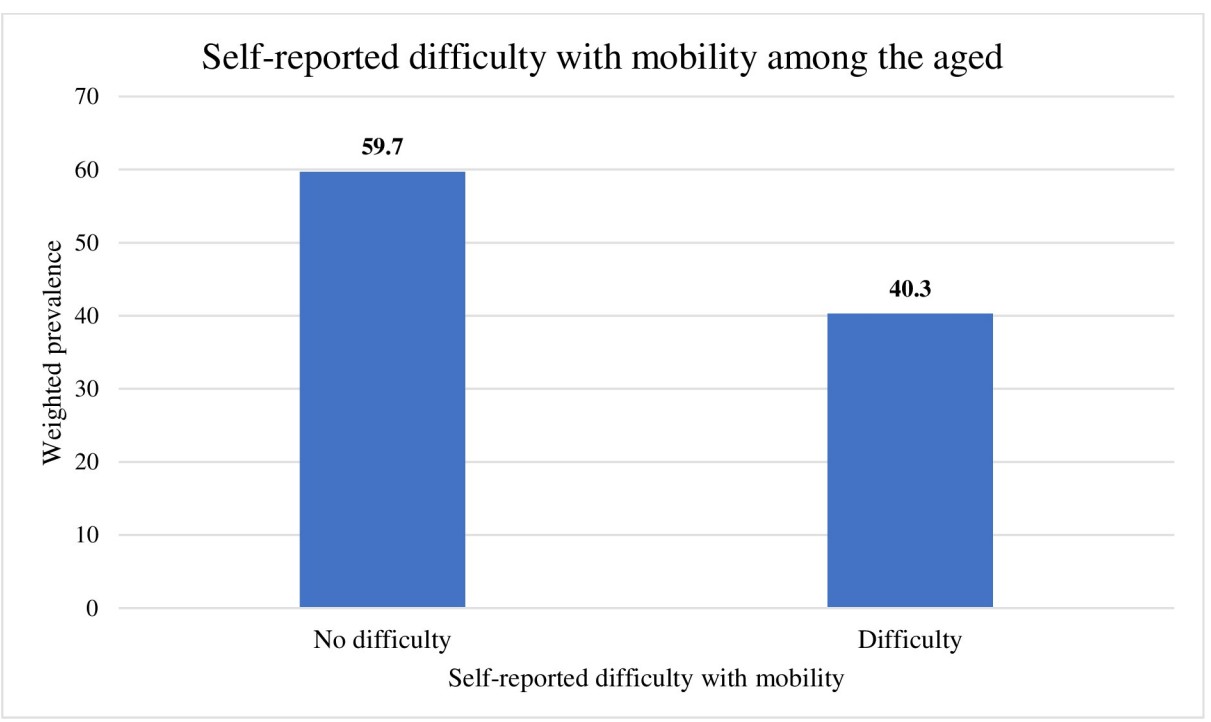

**Fig 1. Self-reported difficulty with mobility among the aged in Ghana.**

Table 3 presents the multi-level logistic regression results of difficulty with mobility among the aged. Participants aged 80 and above compared to 50–59 years [OR = 9.04, 95%CI = 6.07–13.46], females compared to males [OR = 1.39, 95%CI = 1.13–1.72], the widowed compared to those never married [OR = 3.56, 95%CI = 1.86–6.80] and those who had formal education compared to those who never had formal education [OR = 2.13, 95%CI = 1.71–2.65] and participants who had severe/extreme difficulty with household activities [OR = 78.00, 95% CI = 33.72–99.06] had higher odds of difficulty with mobility. Additionally, participants who experienced severe/extreme bodily pains compared to no pains [OR = 24.98, 95%CI = 14.55–42.87], participants who had severe/extreme difficulty sleep problem compared with no sleep problem [OR = 11.40, 95%CI = 5.76–22.55], participants who had difficulty with vision/sight compared with no difficulty [OR = 4.61, 95%CI = 3.71–5.73] and those who do not engaged in vigorous activities compared to those who do [OR = 2.35, 95%CI = 1.84–3.00] had higher odds of difficulty with mobility.

In the adjusted model (Model II), participants aged 80 and above compared to 50–59 years [AOR = 3.05, 95%CI = 1.78–5.22], participants who self-reported poor/bad health status compared to good health [AOR = 2.39, 95%CI = 1.35–4.23], and participants with severe/extreme difficulty performing household activities compared to no difficulty [AOR = 25.12, 95% CI = 11.49–44.91] had higher odds of difficulty with mobility. Additionally, participants who experienced severe/extreme bodily pains compared to no pains [AOR = 4.56, 95%CI = 2.16–9.64], participants who had severe/extreme sleep problems compared to no sleep problem [AOR = 4.15, 95%CI = 1.68–10.29], and participants who had difficulty with sight/vision compared to no difficulty [AOR = 1.56, 95%CI = 1.16–2.10] had higher odds of difficulty with mobility.

**Table 2. Bivariate analysis of factors associated with difficulty with mobility.**

| Variables | Weighted | | Difficulty with Mobility | | X² | P-value |
|---|---|---|---|---|---|---|
| | (n) | (%) | No difficulty, n (%) | Difficulty, n (%) | | |
| **Age group (years)** | | | | | 19.701 | <0.001 |
| 50–59 | 458 | 39.6 | 340 (27.8) | 118 (11.8) | | |
| 60–69 | 680 | 31.8 | 427 (20.4) | 253 (11.4) | | |
| 70–79 | 485 | 19.1 | 218 (8.7) | 267 (10.4) | | |
| ≥80 | 233 | 9.5 | 72 (2.8) | 161 (6.7) | | |
| **Place of residence** | | | | | 0.232 | 0.631 |
| Urban | 765 | 47.1 | 326 (27.7) | 439 (19.4) | | |
| Rural | 1,091 | 52.9 | 618 (20.9) | 473 (32.0) | | |
| **Gender** | | | | | 6.455 | 0.012 |
| Male | 958 | 56.1 | 573 (36.3) | 385 (19.8) | | |
| Female | 898 | 43.9 | 484 (23.4) | 414 (20.5) | | |
| **Marital status** | | | | | 14.058 | <0.001 |
| Never married | 62 | 3.0 | 46 (2.4) | 16 (0.6) | | |
| Married | 985 | 60.2 | 602 (38.4) | 383 (21.8) | | |
| Separated/divorced | 234 | 11.8 | 138 (7.2) | 96 (4.6) | | |
| Widowed | 575 | 25.0 | 271 (11.7) | 304 (13.3) | | |
| **Had formal education** | | | | | 11.048 | <0.001 |
| No | 880 | 38.9 | 430 (19.8) | 450 (19.0) | | |
| Yes | 976 | 61.1 | 627 (39.9) | 349 (21.2) | | |
| **Ethnicity** | | | | | 2.550 | 0.046 |
| Akan | 937 | 46.9 | 539 (29.4) | 398 (17.6) | | |
| Ewe | 131 | 11.0 | 81 (5.7) | 50 (5.3) | | |
| Ga-adangbe | 215 | 13.1 | 137 (8.6) | 78 (4.5) | | |
| Gruma/Grusi/Guan | 129 | 6.2 | 89 (4.4) | 40 (1.8) | | |
| Mande/Mole-dagbani | 444 | 22.8 | 211 (11.7) | 233 (11.0) | | |
| **Religion** | | | | | 0.720 | 0.532 |
| Christian | 1,359 | 75.3 | 34 (1.6) | 28 (1.4) | | |
| Moslem | 329 | 17.1 | 775 (45.0) | 584 (30.4) | | |
| Traditionalist | 106 | 4.6 | 192 (10.7) | 137 (6.4) | | |
| No religion | 62 | 3.1 | 56 (2.4) | 50 (2.1) | | |
| **Have ever worked** | | | | | 0.007 | 0.936 |
| No | 54 | 3.2 | 30 (1.9) | 24 (1.3) | | |
| Yes | 1,802 | 96.8 | 1027 (57.8) | 775 (39.0) | | |
| **Self-rated health status** | | | | | 25.753 | <0.001 |
| Poor/bad | 188 | 10.7 | 30 (2.5) | 158 (8.2) | | |
| Good | 1,668 | 89.3 | 1,027 (57.2) | 641 (32.1) | | |
| **Difficulty with household activities** | | | | | 52.427 | <0.001 |
| No difficulty | 706 | 41.2 | 640 (37.0) | 66 (4.2) | | |
| Mild | 480 | 24.8 | 281 (15.0) | 199 (9.8) | | |
| Moderate | 547 | 28.3 | 124 (7.2) | 423 (21.1) | | |
| Severe/extreme | 123 | 5.7 | 12 (0.6) | 111 (5.1) | | |
| **Bodily pains** | | | | | 37.325 | <0.001 |
| None | 690 | 39.7 | 578 (33.5) | 112 (6.2) | | |
| Mild | 612 | 31.0 | 312 (17.2) | 300 (13.7) | | |
| Moderate | 441 | 22.7 | 145 (7.6) | 296 (15.2) | | |
| Severe/extreme | 113 | 6.6 | 22 (1.4) | 91 (5.2) | | |

*(Continued)*

**Table 2.** (Continued)

| Variables | Weighted | | Difficulty with Mobility | | X$^2$ | P-value |
|---|---|---|---|---|---|---|
| | (n) | (%) | No difficulty, n (%) | Difficulty, n (%) | | |
| **Sleep problems** | | | | | 42.181 | <0.001 |
| None | 1107 | 60.7 | 837 (47.2) | 270 (13.5) | | |
| Mild | 486 | 58.7 | 147 (8.0) | 339 (16.6) | | |
| Moderate | 209 | 11.0 | 61 (3.8) | 148 (7.2) | | |
| Severe/extreme | 54 | 3.6 | 12 (0.7) | 42 (2.9) | | |
| **Difficulty with sight** | | | | | 37.472 | <0.001 |
| No difficulty | 1,084 | 60.4 | 778 (43.5) | 306 (16.9) | | |
| Difficulty | 772 | 39.6 | 279 (16.2) | 493 (23.4) | | |
| **BMI** | | | | | 1.498 | 0.224 |
| Underweight | 997 | 46.2 | 526 (25.9) | 471 (20.3) | | |
| Normal/healthy | 697 | 42.6 | 438 (26.9) | 259 (15.8) | | |
| Overweight | 125 | 8.0 | 77 (5.5) | 48 (2.5) | | |
| Obese | 37 | 3.2 | 16 (1.5) | 21 (1.7) | | |
| **Ever smoked tobacco** | | | | | 0.787 | 0.376 |
| No | 1,698 | 89.4 | 971 (54.0) | 727 (35.4) | | |
| Yes | 158 | 10.6 | 86 (5.7) | 72 (4.9) | | |
| **Ever drink alcohol** | | | | | 2.698 | 0.102 |
| No | 1235 | 59.7 | 688 (35.7) | 369 (24.0) | | |
| Yes | 621 | 40.3 | 547 (26.7) | 252 (13.6) | | |
| **Engage in vigorous work/activities** | | | | | 6.719 | 0.010 |
| No | 1,333 | 69.9 | 691 (38.8) | 642 (31.1) | | |
| Yes | 523 | 30.1 | 366 (20.9) | 157 (9.2) | | |
| **Diagnosed of arthritis** | | | | | 1.037 | 0.309 |
| No | 1,666 | 88.2 | 964 (53.4) | 702 (34.8) | | |
| Yes | 190 | 11.8 | 93 (6.3) | 97 (5.5) | | |
| **Diagnosed of depression** | | | | | 1.269 | 0.261 |
| No | 1,837 | 99.1 | 9 (0.4) | 10 (0.5) | | |
| Yes | 19 | 0.9 | 1,048 (59.3) | 789 (39.8) | | |
| **Had road traffic accident** | | | | | 4.226 | 0.041 |
| No | 1,826 | 98.5 | 1042 (59.1) | 784 (39.4) | | |
| Yes | 30 | 1.5 | 15 (0.6) | 15 (0.9) | | |

## Discussion

We investigated the difficulty with mobility among the aged (≥50 years) in Ghana. The key findings were that among the aged in Ghana, more than one-third (40.3%) had difficulty with mobility. Moreover, difficulty with mobility among the aged in Ghana is associated with age (≥80), self-reported poor health, having severe/extreme difficulty performing household activities, experienced severe/extreme bodily pains, severe/extreme sleep problems, and participants with difficulty with vision.

A global study by Webber et al. revealed that about half of people aged 65 years or older reported difficulties with respect to walking or climbing stairs [23]. Other studies conducted among the aged in other parts of Africa however reported lesser prevalence. For instance, a study conducted among the aged (≥50 years) in Nigeria revealed that 35.8% had difficulty with mobility [24] indicating that nearly two-thirds (64.2%) had no difficulty with mobility. Also, Yaya and colleagues in their cross-national study in South Africa and Uganda revealed

**Table 3. Multilevel mixed-effects logistic regression of factors associated with difficulty with mobility.**

| Variable | Model 0 | | Model I | | Model II | |
|---|---|---|---|---|---|---|
| | OR | 95%CI | OR | 95%CI | AOR | 95%CI |
| **Age group** | | | | | | |
| 50–59 (ref) | | | 1 | 1 | 1 | 1 |
| 60–69 | | | 1.90*** | [1.43–2.54] | 1.75** | [1.21–2.51] |
| 70–79 | | | 4.34*** | [3.18–5.94] | 2.31*** | [1.53–3.49] |
| ≥80 | | | 9.04*** | [6.07–13.46] | 3.05*** | [1.78–5.22] |
| **Gender** | | | | | | |
| Male (ref) | | | 1 | 1 | 1 | 1 |
| Female | | | 1.39** | [1.13–1.72] | 1.31 | [0.92–1.87] |
| **Marital status** | | | | | | |
| Never married (ref) | | | 1 | 1 | 1 | 1 |
| Married | | | 1.75 | [0.93–3.31] | 0.90 | [0.40–2.04] |
| Separated/divorced | | | 2.14* | [1.09–4.23] | 0.73 | [0.31–1.74] |
| Widowed | | | 3.56*** | [1.86–6.80] | 0.80 | [0.36–1.82] |
| **Had formal education** | | | | | | |
| Yes (ref) | | | 1 | 1 | 1 | 1 |
| No | | | 2.13*** | [1.71–2.65] | 0.99 | [0.74–1.32] |
| **Health status** | | | | | | |
| Good (ref) | | | 1 | 1 | 1 | 1 |
| Poor/bad | | | 11.90*** | [7.57–18.65] | 2.39** | [1.35–4.23] |
| **Difficulty with household activities** | | | | | | |
| None (ref) | | | 1 | 1 | 1 | 1 |
| Mild | | | 7.34*** | [5.21–10.33] | 3.69*** | [2.52–5.40] |
| Moderate | | | 42.09*** | [29.09–60.88] | 19.18** | [12.90–28.53] |
| Severe/extreme | | | 78.00*** | [33.72–99.06] | 25.12*** | [11.49–44.91] |
| **Experienced bodily pains** | | | | | | |
| None (ref) | | | 1 | 1 | 1 | 1 |
| Mild | | | 4.84*** | [3.67–6.38] | 2.19*** | [1.53–3.14] |
| Moderate | | | 11.08*** | [8.18–15.02] | 2.84*** | [1.91–4.23] |
| Severe/extreme | | | 24.98*** | [14.55–42.87] | 4.56*** | [2.16–9.64] |
| **Sleep problems** | | | | | | |
| None (ref) | | | 1 | 1 | 1 | 1 |
| Mild | | | 7.12*** | [5.54–9.16] | 3.33*** | [2.41–4.61] |
| Moderate | | | 7.73*** | [5.48–10.90] | 1.87** | [1.18–2.95] |
| Severe/extreme | | | 11.40*** | [5.76–22.55] | 4.15** | [1.68–10.29] |
| **Difficulty with sight/vision** | | | | | | |
| No difficulty (ref) | | | 1 | 1 | 1 | 1 |
| Difficulty | | | 4.61*** | [3.71–5.73] | 1.56** | [1.16–2.10] |
| **Engage in vigorous activity** | | | | | | |
| Yes (ref) | | | 1 | 1 | 1 | 1 |
| No | | | 2.35*** | [1.84–3.00] | 0.86 | [0.62–1.19] |
| **Had road traffic accident** | | | | | | |
| No (ref) | | | 1 | 1 | - | - |
| Yes | | | 1.08 | [0.49–2.37] | - | - |
| **Random effect** | | | | | | |
| ICC | 0.15 | | | | 0.10 | |
| LR test | 66.7*** | | | | 10.29*** | |

*(Continued)*

**Table 3.** (Continued)

| Variable | Model 0 | | Model I | | Model II | |
|---|---|---|---|---|---|---|
| | OR | 95%CI | OR | 95%CI | AOR | 95%CI |
| **Model diagnostics** | | | | | | |
| AIC | 2474.32 | | | | 1546.80 | |
| BIC | 2485.37 | | | | 1668.38 | |
| N | 1,856 | | | | 1,856 | |

95%CI = 95% confidence intervals in brackets, OR = Odds Ratio, AOR = Adjusted Odds Ratio

\* $p < 0.05$

\*\* $p < 0.01$

\*\*\* $p < 0.001$, 1 = Reference category.

that 20% of the aged had difficulty with mobility [25]. However, this study revealed that more than one-third (40.3%) had difficulty with mobility. The likely justification for the difficulty with mobility found in this study is the loss of strength and function that is characterised by sarcopenia in adulthood [26] leading to slow walking, less stability, inefficiency, poor timing and coordination of postures and poor gaiting [27]. Another plausible explanation is the alteration of muscular strength during old age [26] and decrease in physical strength each year after sixty years [26]. Despite the prevalence of difficulty with mobility reported across studies, variations exist in the prevalence estimates which could be attributed to the differences in study settings, sample size, genetic make-up and study designs employed.

This study revealed that participants aged 80 years and above had increased odds of difficulty with mobility. It is logical to argue that mobility difficulties increases with ageing stemming from the deterioration in functioning and degenerative conditions owing to sarcopenia [28]. This argument is supported by research in Ghana that revealed that the aged are mostly confronted with degenerative conditions and physical deterioration that affect their capacity to effectively function including mobility [29]. Even though, mobility challenges could be prevalent at younger ages, research revealed that mobility difficulties are mostly prevalent at old age which could also be attributed to the decline in strength in the muscles [30]. A research conducted in Nigeria also revealed that the aged (>70 years) are at heightened risk of mobility difficulties [24]. This decline in mobility could be present even in the absence of co-morbidity because ageing induces biological and functional decline at several levels (including loss of muscle strength and mass and decline in balance) resulting in difficulty in mobility [26].

The study revealed that participants who had severe/extreme difficulty performing household activities had difficulties with mobility. This is because the age-related decline in capabilities and functions may inhibit or compromise older adults' ability to perform household activities. Corroborating and affirming the above finding, a study in Taiwan revealed that engaging in household activities was associated with the probability of survival, and respondents with difficulty engaging in household activities had mobility limitations [31]. Similarly, in Finland, research revealed that persons having problems conducting daily activities had mobility difficulties [32]. In the USA, research revealed that the aged with problems conducting home and daily activities were more likely to have mobility difficulties [33]. Engaging in household activities makes the elderly physically active, and more productive thereby increasing mobility [31].

The study indicated that participants who had severe/extreme sleep problems had higher odds of difficulty with mobility. Corroborating and affirming the study, a cross-sectional study conducted in Finland among the aged revealed that having sleeping disorders or insomnia was

positively linked to mobility limitations in both men and women aged 55 and above [32]. Consistent with the findings, a research conducted in the USA also confirmed that having fragmented sleep was more associated with functional limitations including difficulty with mobility for the aged [34]. This fragmented or difficulty sleeping occurs chronically among the older populations [32], and because ageing is linked to a decline in physical performance which often translates into physical disability and loss of independence, concomitant sleep problems may worsen or exacerbate the age-related decline in physical function [32], leading to difficulty in mobility.

The study also revealed that participants who experienced severe/extreme bodily pains had increased odds of difficulty with mobility. In a related study in West Africa, health conditions including bodily pains and discomfort are risk factors for mobility disability [30]. Consistent with the above finding, in USA, research found that the presence of mild-to-moderate pain among the aged was independently associated with poor or difficulty with mobility among the aged [35]. The association between severe/extreme bodily pain and mobility limitation could however be a result of reverse causation. Musich et al. for instance, in their study in the USA found that higher levels of mobility limitations were linked to pain among the aged. The authors, therefore, recommended mobility-enhancing interventions that could promote successful ageing [36].

Participants who had severe/extreme difficulty with vision had higher odds of difficulty with mobility. Similarly, in a study in rural India, the authors found mobility losses of 5.1, 10.2, and 23.4 points of 100 among older people with visual difficulties, low vision, and blindness, respectively [37]. This finding is also similar to other related studies conducted in high-income countries. For instance, related studies in Israel, Canada, China, and USA found that the aged who had visual difficulty had walking and stair-climbing difficulties [38–41]. Research revealed that older adults with vision difficulties had significantly difficulty with mobility including slower walking speeds compared to non-visually impairment individuals [42]. The likely explanation is that the slowness in walking speed is a way to maintain or improve their mobility safety.

Additionally, the study revealed that the aged who reported having poor health status compared with good health had higher odds of difficulty with mobility. Poor health has been shown to be a determinant of mobility limitations among the aged in a similar study in Nigeria [24]. In other words, the aged who perceived to have good health (that is, devoid of cardiovascular and metabolic diseases, osteoporosis, muscular weakness) had decreased odds of difficulty with mobility [43]. In congruence with the above, a study published elsewhere revealed that difficulties with mobility are central to ageing [44]. The authors further justified that having poor health is associated with limitations in both walking and driving. Also, research in the United Kingdom revealed that even older adults who perceived their health to be good are not devoid of mobility limitations due to ageing [45]. However, among these healthy older people, mobility limitation is due to weakness of the body and not their health status [45].

## Strength and limitations

The study is novel because it presented updated information on difficulty with mobility among the aged in Ghana. Also, the study utilised secondary data from a nationally representative cross-sectional survey with a relatively large sample size (n = 1,856). Additionally, the researcher applied rigorous, advanced statistical and analytical methods to analyse the data for the study making it robust. However, despite the above-mentioned strengths, the study had its limitations that cannot be overemphasised. The study utilised a dataset that was collected from respondents' self-report making recall bias inevitable. Additionally, the study considered the

aged population as individuals 50 years or older which does not correspond with the definition of the older population by the Ghana Statistical Service ($\geq$60 years) which could translate into a difference in the outcome (mobility) measured. The study is also liable to social desirability biases (tendency of respondents to bias responses to make them appropriate or socially acceptable) due to the cross-sectional nature of SAGE study. The cross-sectional nature of the survey also led to the failure to establish a causal relationship. Furthermore, some of the associations found in this study could be due to reverse causality.

## Conclusion

The aged in Ghana had relatively higher prevalence (40.3%) of difficulty with mobility which is influenced by demographics, health status, and degeneration associated with ageing. This highlights the need to provide social support and strengthen social capital for the aged in Ghana, especially those with difficulty with vision, experiencing bodily pains and had poor health status. Additionally, the Government and stakeholders should provide assistive devices for the aged and geriatric care including recreational fields and care homes to address the health and physical needs of the aged in Ghana.

### Policy implications, practice, and research

The government should develop and implement health policies that would address the needs of older adults with mobility difficulties. Additionally, the government should ensure public spaces, buildings, and transportation systems are easily accessible and geriatric friendly Health providers should provide specialised geriatric health services, including physical activity, therapy and rehabilitation programs to support the mobility and independence of the aged. Future studies must investigate the long-term effects of difficulty with mobility and quality of life among the aged.

## Supporting information

**S1 Table. Individual questionnaire and coding.**
(PDF)

**S2 Table. Multi-collinearity test results.**
(PDF)

## Acknowledgments

We are grateful to all participants and the interviewers who made the SAGE study in Ghana possible. We are also grateful to the Ministry of Health, Ghana, the University of Ghana's Department of Community Health, and the Ghana Statistical Service for the contributions and assistance provided. Finally, we are most grateful to the WHO SAGE team for granting us access to use the dataset.

## Author Contributions

**Conceptualization:** Kingsley Boakye, Daniel Boateng, Emmanuel Kweku Nakua.

**Data curation:** Kingsley Boakye.

**Formal analysis:** Kingsley Boakye.

**Methodology:** Kingsley Boakye, Antoinette Ama Aidoo, Daniel Boateng.

**Supervision:** Mohammed Aliyu, Daniel Boateng, Emmanuel Kweku Nakua.

**Writing – original draft:** Kingsley Boakye, Antoinette Ama Aidoo, Mohammed Aliyu, Daniel Boateng, Emmanuel Kweku Nakua.

**Writing – review & editing:** Kingsley Boakye, Antoinette Ama Aidoo, Mohammed Aliyu, Daniel Boateng, Emmanuel Kweku Nakua.

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
