## [Decision Letter · Decision Letter 0]

11 Dec 2023

PONE-D-23-22220Difficulty with Mobility among the aged in Ghana: Evidence from Wave 2 of the World Health Organization’s study on Global Ageing and Adult HealthPLOS ONE

Dear Dr. Kingsley,

Thank you for submitting your manuscript to PLOS ONE. After careful consideration, we feel that it has merit but does not fully meet PLOS ONE’s publication criteria as it currently stands. Therefore, we invite you to submit a revised version of the manuscript that addresses the points raised during the review process. Please submit your revised manuscript by Jan 25 2024 11:59PM. If you will need more time than this to complete your revisions, please reply to this message or contact the journal office at plosone@plos.org. Please include the following items when submitting your revised manuscript:A rebuttal letter that responds to each point raised by the academic editor and reviewer(s). You should upload this letter as a separate file labeled 'Response to Reviewers'.A marked-up copy of your manuscript that highlights changes made to the original version. You should upload this as a separate file labeled 'Revised Manuscript with Track Changes'.An unmarked version of your revised paper without tracked changes. You should upload this as a separate file labeled 'Manuscript'.

We look forward to receiving your revised manuscript.

Kind regards,

Kahsu Gebrekidan

Academic Editor

PLOS ONE

Journal Requirements:

Reviewers' comments:

Reviewer's Responses to Questions

**Comments to the Author**

1. Is the manuscript technically sound, and do the data support the conclusions?

Reviewer #1: Partly

Reviewer #2: Yes

2. Has the statistical analysis been performed appropriately and rigorously? 

Reviewer #1: No

Reviewer #2: N/A

3. Have the authors made all data underlying the findings in their manuscript fully available?

Reviewer #1: Yes

Reviewer #2: Yes

4. Is the manuscript presented in an intelligible fashion and written in standard English?

Reviewer #1: Yes

Reviewer #2: Yes

5. Review Comments to the Author

Reviewer #1: Research idea to generate evidence for geriatric care is good. Authors have skills to do secondary data analysis. However, there are conceptual issues in the analysis:

1. Dependent variable is construed by merging two questions: one on mobility difficulty and the other in vigorous activity. This is likely to inflate the mobility problem. It is not expected that elderlies will not face problems for the vigorous activities. Therefore, authors should either drop the second variable or do separate analysis for the same. Secondly it is not best idea to merge mild problem with severe problem and treat in one category. Oversimplification of data set for analysis sometimes lead to loss of information. Mild problems can be merged with no problems also. Or some scoring system can be devised. It will be interesting to know how these problems increase with age. ( including the severity)

2. Predictors also need relook. Pains and discomfort can be merged into one predictor. Vigorous activity is also included in predictor and it was also there in the dependent variable. This may be dropped from predictor. Authors can also analysis association with smoking and alcohol etc for which data is available to them. Stroke will lead to mobility problem. It can be dropped as predictor in study related to ageing.

3. Authors to look to typo errors. Data table for self-reported health problem is misreported.

4. Collinearity : How have authors tested collinearity for categorical predictors and what cutoff of VIF was used for decision.

Reviewer #2: Abstract and background well-articulated. Methods as a title is required in line 107, before describing the data collection and analysis.

Dependent variables

The quotes required for the responses should not be doubled. I would suggest revising and using the single quote.

Independent variables

The sentence is too long to list 18 independent variables in one sentence. Please try to split and describe them with conjunction. Similar comment from line 137-145, it is a long sentence. In addition, single quotes are required for the possible responses.

Results

I would suggest describing the number and per cent: N (%). It is confusing for the reader that nearly 1/3, 2/3, ¾, nearly half… in all your description. The author should be selective in describing the key demographics. It is not necessary to describe all variables in the table. Similar comment to the description in figure 1.

The description in table 2 and table 3 is not clear, and a bit confusing. The author simply listed and compared the variables. What is the result indicating, what does the association mean? All this should be revised.

Discussion

The first two paragraphs are repeating the results. In discussion, the author is expected to interpret the results. I also have seen some irrelevant words such as ‘interestingly’ in your discussion. Please try to revise and discuss the key findings of your study.

Declaration

Why is ethical approval written twice? If this is the place to write, it is not necessary to mention in the method section.

Why is not applicable consent for publication?

References

The list of references should be consistent and well written. Please have a look at references such as reference number 20, 27 and the last references.

6. PLOS authors have the option to publish the peer review history of their article (what does this mean?). If published, this will include your full peer review and any attached files.

Reviewer #1: No

Reviewer #2: No

---

## [Author Response · Author response to Decision Letter 0]

10 Jan 2024

Please, I have addressed all the comments raised in the revised manuscript appropriately

---

## [Decision Letter · Decision Letter 1]

9 Apr 2024

PONE-D-23-22220R1Difficulty with mobility among the aged in Ghana: Evidence from Wave 2 of the World Health Organization’s Study on Global Ageing and Adult HealthPLOS ONE

Dear Dr. Boakye,

Thank you for submitting your manuscript to PLOS ONE. After careful consideration, we feel that it has merit but does not fully meet PLOS ONE’s publication criteria as it currently stands. Therefore, we invite you to submit a revised version of the manuscript that addresses the points raised during the review process.

We look forward to receiving your revised manuscript.

Kind regards,

Kahsu Gebrekidan

Academic Editor

PLOS ONE

Journal Requirements:

Reviewers' comments:

Reviewer's Responses to Questions

**Comments to the Author**

1. If the authors have adequately addressed your comments raised in a previous round of review and you feel that this manuscript is now acceptable for publication, you may indicate that here to bypass the “Comments to the Author” section, enter your conflict of interest statement in the “Confidential to Editor” section, and submit your "Accept" recommendation.

Reviewer #2: (No Response)

Reviewer #3: All comments have been addressed

2. Is the manuscript technically sound, and do the data support the conclusions?

Reviewer #2: Yes

Reviewer #3: Yes

3. Has the statistical analysis been performed appropriately and rigorously? 

Reviewer #2: (No Response)

Reviewer #3: Yes

4. Have the authors made all data underlying the findings in their manuscript fully available?

Reviewer #2: Yes

Reviewer #3: Yes

5. Is the manuscript presented in an intelligible fashion and written in standard English?

Reviewer #2: Yes

Reviewer #3: Yes

6. Review Comments to the Author

Reviewer #2: Abstract

The first sentences of the abstract should be revised. The flow of the sentence is not clear. I would suggest starting from the second sentence.

Methods

What is the standard age for the aged population? Why do the authors include participants aged 50 years and over? What does the aged in Ghana mean?

Conclusion: It is a repetition of results especially the first sentences. It will be more relevant to include the summary of the findings, the implication and recommendation.

Dependent variables

As I mentioned in the previous comments, I have seen any change with the quotes. I believe a single quote should be used for the responses. The authors said it is revised. However, I have not seen any revised version here.

Independent variables

Similar comments with the quotes to the independent variables starting from line 144 to 155. Here the author revised the long sentences which are important. However, the author used additionally as a conjunction round three times. It will be more relevant to use synonyms rather than repeating the same word.

Results

The description of results is not revised. The Authors simply describe one third, nearly… repeatedly. It is relevant to use different types of description rather than repetition of words. I would again suggest the number and percent N (%). Similar comments to all types of description.

Reviewer #3: The manuscript has been suitably updated as advised by the reviewers. SAGE is a benchmark longitudinal study in the field of ageing in community.

7. PLOS authors have the option to publish the peer review history of their article (what does this mean?). If published, this will include your full peer review and any attached files.

Reviewer #2: No

Reviewer #3: No

---

## [Author Response · Author response to Decision Letter 1]

17 Jul 2024

Please, I have addressed all the comments raised by the reviewers in the response to reviewers document

---

## [Decision Letter · Decision Letter 2]

12 Aug 2024

Difficulty with mobility among the aged in Ghana: Evidence from Wave 2 of the World Health Organization’s Study on Global Ageing and Adult Health

PONE-D-23-22220R2

Dear Mr. Kingsley,

We’re pleased to inform you that your manuscript has been judged scientifically suitable for publication and will be formally accepted for publication once it meets all outstanding technical requirements.

Kind regards,

Kahsu Gebrekidan, Ph.D.

Academic Editor

PLOS ONE

Additional Editor Comments (optional):

Reviewers' comments:

Reviewer's Responses to Questions

**Comments to the Author**

1. If the authors have adequately addressed your comments raised in a previous round of review and you feel that this manuscript is now acceptable for publication, you may indicate that here to bypass the “Comments to the Author” section, enter your conflict of interest statement in the “Confidential to Editor” section, and submit your "Accept" recommendation.

Reviewer #2: All comments have been addressed

2. Is the manuscript technically sound, and do the data support the conclusions?

Reviewer #2: Yes

3. Has the statistical analysis been performed appropriately and rigorously? 

Reviewer #2: N/A

4. Have the authors made all data underlying the findings in their manuscript fully available?

Reviewer #2: Yes

5. Is the manuscript presented in an intelligible fashion and written in standard English?

Reviewer #2: Yes

6. Review Comments to the Author

Reviewer #2: The manuscript is well revised. I have a few suggestions and a question.

Result

The description in table three from line 213- 235 is not understood. It is difficult to follow the sentence. It is too long.

Discussion

I would suggest for the second paragraph of the discussion (251-265) to interpret the results in the discussion rather than only comparing with other studies conducted and justifying the difference.

The author used ‘study revealed that participants… repeatedly. Please use synonyms. In line 287. Is the study conducted in the USA consistent with your study or its contrast? There is no conjunction word with your study.

7. PLOS authors have the option to publish the peer review history of their article (what does this mean?). If published, this will include your full peer review and any attached files.

Reviewer #2: No

---

## [Editor Report · Acceptance letter]

16 Aug 2024

PONE-D-23-22220R2 

PLOS ONE

Dear Dr. Boakye, 

I'm pleased to inform you that your manuscript has been deemed suitable for publication in PLOS ONE. Congratulations! Your manuscript is now being handed over to our production team.

Kind regards, 

on behalf of

Dr. Kahsu Gebrekidan 

Academic Editor

PLOS ONE